# Command Filter-based Adaptive Fuzzy Tracking Control for Intelligent Ship Autopilot with Full-state Constraints

Lingjia Zhao
*The School of Mathematical Sciences*
*Bohai University*
Jinzhou, Liaoning, China
e-mail: lingjiazhao7@163.com

Dewen Tong
*Jinzhou No.4 Middle School*
Jinzhou, Liaoning, China
e-mail: 15542815157@126.com

Huanqing Wang
*The School of Mathematical Sciences*
*Bohai University*
Jinzhou, Liaoning, China
e-mail: ndwhq@163.com

*Abstract*—In this article, a command-filtered composite adaptive fuzzy tracking control problem is considered for intelligent ship autopilot with full-state constraints. Fuzzy logic systems (FLSs) are utilized to tackle the unknown nonlinear functions. By the method of inverse mapping to a tan-type function, the issue of full-state constraints for nonlinear systems is disposed. Through the use of serial-parallel estimation model (SPEM), the forecast bias and the track bias can change the weights of FLSs and the approximate characteristics of FLSs will be improved. Then, by constructing the command filter, the issue of complex explosion will be effectively solved. The designed control strategy can ensure the boundedness of all states in the considered system satisfies the state constraints. The output also can track the desire trajectory. Finally, the simulation example demonstrates the validity of the proposed control scheme.

*Index Terms*—Command filter, fuzzy logic systems, composite adaptive control, full-state constraints, backstepping

## I. INTRODUCTION

Adaptive backstepping technique and Lyapunov analysis method were broadly utilized to handle the control issue of nonlinear systems. However, the aforesaid approaches can not settle the problem of unknown nonlinear terms in the considered system. Radial basis function neural networks or FLSs were utilized to approximate the unknown nonlinear function in the system. In the last decades, by using the backstepping technique, a lot of fuzzy control strategies were proposed, such as [1]- [3]. Subsequently, the SPEM in many articles were proposed which were widely used to improve the approximation of neural networks or FLSs, such as [4]-[6]. However, in the process of controller design, it was found that repeated derivation of virtual controller would cause a complexity explosion problem. The command-filter approach is developed in [7] to tackle the issue. It not only overcame the computational complexity problem, but also compensated for the filtering errors. In addition, due to physical environment, security factors and other reasons, many practical systems usually operate under full state constraints. Therefore, the issue of the nonlinear systems control with full-states constrains is a challenge and the tan-type nonlinear mapping approach [8] can settle it. In this work, by utilizing the adaptive

backstepping control technique, a composite adaptive fuzzy control strategy is designed for intelligent ship autopilot with full-state constraints. The full-state constraints are handled by introducing the tan-type nonlinear mapping approach. By utilizing the command filter, the computational complexity problem is tackled. By combining the Lyapunov stability theorem, we can attest that all states in the considered system satisfy the state constraints. The output signal also can track the reference signals. This article can be developed according to the following structure. In section II, the propaedeutics and problem formulation are offered. Section III and section IV show the design of controller and the result of analysis, respectively. Simulation result verifies the validity of the strategy we proposed in section V. The conclusions are given in section VI.

## II. PRELIMINARIES AND PROBLEM FORMULATIONS

Considered the following ship heading control mathematical model:

$$\ddot{\phi} + \frac{1}{T}H(\dot{\phi}) = \frac{K}{T}\delta \tag{1}$$

where $\phi$ and $\delta$ denote the ship heading angle and the rudder angle, respectively. $\delta$ is the control input, $K$ indicates the rudder gain, $T$ stands for the time constant. The unknown nonlinear function $H(\dot{\phi})$ is a real-valued constants and is expressed as follows: $H(\dot{\phi}) = a_1\dot{\phi} + a_2\dot{\phi}^3 + a_3\dot{\phi}^5 + \ldots, a_i$, for $i = 1, 2, 3, \ldots$

Defining $x_1 = \phi, x_2 = \dot{\phi}$ and $u = \frac{K\delta}{T}$, we can transform (1) into the following form:

$$\begin{cases} \dot{x}_1 = x_2 \\ \dot{x}_2 = f(x_2) + u \\ y = x_1 \end{cases} \tag{2}$$

where $x_1, x_2$ are the states of (2), $u$ denotes the control input, $y$ represents the output of the system. $f(x_2) = -\frac{1}{T}H(x_2)$ represents an unknown smooth nonlinear function.

*Assumption 1*: [10] The desire ship heading signal $y_r$ and its first-order derivative $\dot{y}_r$ are known and bounded.

*Lemma 1* [11]: For $x, y \in R$, the inequality holds as follows:

$$xy \leq \frac{\epsilon^p}{p}|x|^p + \frac{1}{q\epsilon^q}|y|^q \tag{3}$$

where $\epsilon > 0$, $p > 1$, $q > 1$ and $(p-1)(q-1)$.

*Lemma 2* [12]: For $\forall \sigma > 0$, and a continuous $\Psi(x)$ defined on a compact set $\Omega$, there has a FLS $W^T \phi(x)$ to approximate it, yields

$$\sup_{x \in \Omega} |\Psi(x) - W^T \phi(x)| \leq \sigma \tag{4}$$

where $W = [\omega_1, \omega_2, \cdots, \omega_m]^T$ denotes the desired weight vector. $\Psi(x) = \frac{[l_1(x), l_2(x), \cdots, l_m(x)]^T}{\sum_{i=1}^m \hbar_i(x)}$ represents the basis function vector with $m > 1$ is the rule member, and $\hbar_i(x) = \exp[\frac{-(x-\vartheta_i)^T(x-\vartheta_i)}{\varrho_i^T \varrho_i}]$ is Gaussian function, in which $\vartheta_i = [\vartheta_{i,1}, \vartheta_{i,2}, \cdots, \vartheta_{i,n}]$ denotes the center vector, $\varrho_i = [\varrho_{i,1}, \varrho_{i,2}, \cdots, \varrho_{i,n}]$ represents the width of the Gaussian function.

*Definition 4* [8]: A nonlinear mapping function $\Phi_i$ is described as follows:

$$q_i = \Phi_i(x_i) = \frac{2a_i}{\pi} \tan(\frac{\pi x_i}{2a_i}) \qquad i = 1, \ldots, n \tag{5}$$

where $a_i > 0$ represents a differentiable constraint function of $x_i$. $\Phi_i$ denotes a continuous elementary function.

From (5), the inverse mapping of $\Phi_i$ is expressed as

$$x_i = \Phi_i^{-1} = \frac{2a_i}{\pi} \arctan(\frac{\pi q_i}{2a_i}) \tag{6}$$

The differential of $x_i$ $(i = 1, 2)$ is expressed as

$$\dot{x}_i = p_i(q_i) + \frac{\dot{q}_i}{\kappa_i(q_i)} \tag{7}$$

where

$$\begin{cases} p_i(q_i) = \frac{2\dot{a}_i}{\pi} \arctan(\frac{\pi q_i}{2a_i}) - \frac{q_i \dot{a}_i}{(1+(\frac{\pi q_i}{2a_i})^2)a_i} \\ \kappa_i(q_i) = 1 + (\frac{\pi q_i}{2a_i})^2 \end{cases} \tag{8}$$

Plugging (7) and (8) into (2), we get

$$\begin{cases} \dot{q}_1 = F_1(\bar{q}_2) + q_2 \\ \dot{q}_2 = \kappa_2(q_2)u + F_2(q_2) \\ y^* = q_1 \end{cases} \tag{9}$$

where $\bar{q}_2 = [q_1, q_2]^T$ and

$$\begin{cases} F_1(\bar{q}_2) = (x_2 - p_1(q_1))\kappa_1(q_1) - q_2 \\ F_2(q_2) = (f_2(x_2) - p_2(q_2))\kappa_2(q_2) \end{cases} \tag{10}$$

## III. CONTROLLER DESIGN PROCESS

In this part, an adaptive fuzzy control scheme is designed for the system (9). The coordinate transformations are given as follows:

$$\begin{cases} z_1 = q_1 - y_r^* \\ z_2 = q_2 - \alpha_2^f \end{cases} \tag{11}$$

where $y_r^* = \frac{2a_1}{\pi} \tan(\frac{\pi y_r}{2a_1})$.

In order to deal with computational complexity problem resulted in the repeated derivative of the indirect controller, a first-order filter is designed as [9]

$$\tau_2 \dot{\alpha}_2^f + \alpha_2^f = \alpha_1 \quad \alpha_2^f(0) = \alpha_1(0) \tag{12}$$

where $\alpha_2^f$ is the filter input, $\tau_2$ is a constant. The filter error is expressed as $\eta_2 = \alpha_2^f - \alpha_1$. If inequalities $|\alpha_1| \leq \rho$ and $|\dot{\alpha}_1| \leq \bar{\rho}$ hold with $\rho > 0$ and $\bar{\rho} > 0$ are constants, then for $\forall \bar{\eta}_2 > 0$ is a constant, there exists $\tau_2 > 0$ such that $|\alpha_2^f - \alpha_1| = |\eta_2| \leq \bar{\eta}_2$.

**Step** 1. From (9) and (11), the differential of the tracking error $z_1$ is

$$\begin{aligned} \dot{z}_1 &= \dot{q}_1 - \dot{y}_r^* \\ &= F_1(\bar{q}_2) + q_2 - \dot{y}_r^* \\ &= F_1(\bar{q}_2) + z_2 + \eta_2 + \alpha_1 - \dot{y}_r^* \end{aligned} \tag{13}$$

The error compensation signal designed eliminates the influence of $\alpha_2^f - \alpha_1$. The compensating signals $\xi_1$ is defined as

$$\dot{\xi}_1 = \eta_2 + \xi_2 - c_1 \xi_1 \tag{14}$$

where $c_1$ stands for positive parameter.

The compensated tracking error is designed as $\varpi_1 = z_1 - \xi_1$ and the derivative of $\varpi_1$ is expressed as

$$\dot{\varpi}_1 = \dot{z}_1 - \dot{\xi}_1 = \varpi_2 + \alpha_1 + F_1(\bar{q}_2) - \dot{y}_r^* + c_1 \xi_1 \tag{15}$$

where $\varpi_2 = z_2 - \xi_2$.

Based on Lemma 1, the FLS is employed to approximate uncertain function $F_1(\bar{q}_2)$. $F_1(\bar{q}_2)$ is expressed as the following equality:

$$F_1(\bar{q}_2) = W_1^T \phi_1(\bar{q}_2) + \delta_1(\bar{q}_2) \tag{16}$$

where $\delta_1(\bar{q}_2)$ is the bounded approximation error and satisfies $|\delta_1(\bar{q}_2)| \leq \bar{\delta}_1$.

By plugging (16) into (15), one can obtain

$$\dot{\varpi}_1 = \varpi_2 + \alpha_1 + W_1^T \phi_1(\bar{q}_2) + \delta_1(\bar{q}_2) - \dot{y}_r^* + c_1 \xi_1 \tag{17}$$

Create the function $V_1$ as

$$V_1 = \frac{1}{2}\varpi_1^2 + \frac{1}{2\beta_1}\tilde{W}_1^T \tilde{W}_1 \tag{18}$$

where $\beta_1 > 0$ represents the designed parameter and $\tilde{W}_1 = W_1 - \hat{W}_1$. $\hat{W}_1$ is the estimation of $W_1$, and $\tilde{W}_1$ is the estimation error.

Subsequently, the differential of $V_1$ is

$$\dot{V}_1 = \varpi_1 \dot{\varpi}_1 - \frac{1}{\beta_1}\tilde{W}_1^T \dot{\hat{W}}_1 \tag{19}$$

By plugging (17) into (19), it yields

$$\begin{aligned} \dot{V}_1 = &\varpi_1(\varpi_2 + \alpha_1 + W_1^T \phi_1(\bar{q}_2) + \delta_1(\bar{q}_2) - \dot{y}_r^* + c_1 \xi_1 \\ &+ W_1^T \phi_1(q_1) - W_1^T \phi_1(q_1)) - \frac{1}{\beta_1}\tilde{W}_1^T \dot{\hat{W}}_1 \end{aligned} \tag{20}$$

By applying Lemma 1, the following inequalities hold

$$\varpi_1 \delta_1(\bar{q}_2) \leq \frac{1}{2}\varpi_1^2 + \frac{1}{2}\bar{\delta}_1^2$$
$$\varpi_1(W_1^T \phi_1(\bar{q}_2) - W_1^T \phi_1(q_1)) \leq \varpi_1^2 + \|W_1\|^2 \tag{21}$$

Then, we can obtain

$$\begin{aligned} \dot{V}_1 = &\varpi_1(\frac{3}{2}\varpi_1 + \varpi_2 + \alpha_1 - \dot{y}_r^* + c_1 \xi_1 + \hat{W}_1^T \phi_1(q_1)) \\ &- \frac{1}{\beta_1}\tilde{W}_1^T(\dot{\hat{W}}_1 - \beta_1 \varpi_1 \phi_1(q_1)) + \frac{1}{2}\bar{\delta}_1^2 + \|W_1\|^2 \end{aligned} \tag{22}$$

Next, the virtual controller $\alpha_1$ is designed as follows:

$$\alpha_1 = -\frac{3}{2}\varpi_1 - b_1\varpi_1 - c_1\xi_1 - \hat{W}_1^T\phi_1(q_1) + \dot{y}_r^* \qquad (23)$$

and the adaptive law $\dot{\hat{W}}_1$ is constructed as

$$\dot{\hat{W}}_1 = \beta_1\varpi_1\phi_1(q_1) - \zeta_1\hat{W}_1 \qquad (24)$$

where $b_1$ and $\zeta_1$ are designed positive parameters.

By substituting (23) and (24) into (22), we get

$$\dot{V}_1 \le -b_1\varpi_1^2 + \varpi_1\varpi_2 + \frac{\zeta_1}{\beta_1}\tilde{W}_1^T\hat{W}_1 + \frac{1}{2}\bar{\delta}_1^2 + \|W_1\|^2 \quad (25)$$

**Step** 2. Based on system (9) and the coordinate transformations (11), the time derivative of $z_2$ is

$$\dot{z}_2 = \dot{q}_2 - \dot{\alpha}_2^f = \kappa_2(q_2)u + F_2(q_2) - \dot{\alpha}_2^f \qquad (26)$$

Similar to step 1, the compensating signal $\xi_2$ is constructed as

$$\dot{\xi}_2 = -\xi_1 - c_2\xi_2 \qquad (27)$$

where $c_2$ is a positive constant.

Define the compensated tracking error as $\varpi_2 = z_2 - \xi_2$ and its derivative is

$$\dot{\varpi}_2 = \dot{z}_2 - \dot{\xi}_2 = \kappa_2(q_2)u + F_2(q_2) - \dot{\alpha}_2^f + \xi_1 + c_2\xi_2 \quad (28)$$

The FLS is applied to approximate the unknown function $F_2(q_2)$ such that

$$F_2(q_2) = W_2^T\phi_2(q_2) + \delta_2(q_2) \qquad (29)$$

where $\delta_2(q_2)$ stands for the bounded approximation error and satisfies $\delta_2(q_2) \le \bar{\delta}_2$.

By plugging (29) into (28), one has

$$\begin{aligned}
\dot{\varpi}_2 &= \dot{z}_2 - \dot{\xi}_2 \\
&= \kappa_2(q_2)u + W_2^T\phi_2(q_2) + \delta_2(q_2) \\
&\quad - \dot{\alpha}_2^f + \xi_1 + c_2\xi_2
\end{aligned} \qquad (30)$$

where $k_2$ is a designed parameter.

In order to improve the approximation of FLSs , a composite adaptive update law is designed. Defining error $e_2 = q_2 - \hat{q}_2$ and $\hat{q}_2$ can be gotten from the following SPEM:

$$\dot{\hat{q}}_2 = \kappa_2(q_2)u + \hat{W}_2^T\phi_2(q_2) + k_2e_2 + \frac{1}{2}e_2 \qquad (31)$$

where $k_2$ represents the designed parameter.

By combining (30) with (31), one has

$$\dot{e}_2 = \tilde{W}_2^T\phi_2(q_2) + \delta_2(q_2) - k_2e_2 - \frac{1}{2}e_2 \qquad (32)$$

Construct the Lyapunov function as

$$V = V_1 + \frac{1}{2}\varpi_2^2 + \frac{1}{2\beta_2}\tilde{W}_2^T\tilde{W}_2 + \frac{1}{2}e_2^2 \qquad (33)$$

where $\beta_2$ is a positive constant and $\tilde{W}_2 = W_2 - \hat{W}_2$. $\hat{W}_2$ is the estimation of $W_2$, and $\tilde{W}_2$ is the estimation error.

Then, the $\dot{V}$ is expressed as

$$\begin{aligned}
\dot{V} &= \dot{V}_1 + \varpi_2\dot{\varpi}_2 - \frac{1}{\beta_2}\tilde{W}_2^T\dot{\hat{W}}_2 + e_2\dot{e}_2 \\
&\le -b_1\varpi_1^2 + \varpi_2(\varpi_1 + \kappa_2(q_2)u + \hat{W}_2^T\phi_2(q_2) \\
&\quad + \xi_1 + c_2\xi_2 + \delta_2(q_2) - \dot{\alpha}_2^f) + \frac{\zeta_1}{\beta_1}\tilde{W}_1^T\hat{W}_1 \\
&\quad - \frac{1}{\beta_2}\tilde{W}_2^T(\dot{\hat{W}}_2 - \beta_2(\varpi_2\phi_2(q_2) + e_2\phi_2(q_2))) \\
&\quad + e_2(\delta_2(q_2) - k_2e_2 - \frac{1}{2}e_2) + \frac{1}{2}\bar{\delta}_1^2 + \|W_1\|^2 \quad (34)
\end{aligned}$$

Based on Lemma 1, it yields

$$\begin{aligned}
\varpi_2\delta_2(q_2) &\le \frac{1}{2}\varpi_2^2 + \frac{1}{2}\bar{\delta}_2^2 \\
e_2\delta_2(q_2) &\le \frac{1}{2}e_2^2 + \frac{1}{2}\bar{\delta}_2^2
\end{aligned} \qquad (35)$$

Combining (35) with (34), we get

$$\begin{aligned}
\dot{V} &= \dot{V}_1 + \varpi_2\dot{\varpi}_2 - \frac{1}{\beta_2}\tilde{W}_2^T\dot{\hat{W}}_2 + e_2\dot{e}_2 \\
&\le -b_1\varpi_1^2 + \varpi_2(\varpi_1 + \frac{1}{2}\varpi_2 + \kappa_2(q_2)u + \hat{W}_2^T\phi_2(q_2) \\
&\quad + \xi_1 + c_2\xi_2 - \dot{\alpha}_2^f) + \frac{\zeta_1}{\beta_1}\tilde{W}_1^T\hat{W}_1 \\
&\quad - \frac{1}{\beta_2}\tilde{W}_2^T(\dot{\hat{W}}_2 - \beta_2(\varpi_2\phi_2(q_2) + e_2\phi_2(q_2))) \\
&\quad - k_2e_2^2 + \frac{1}{2}\bar{\delta}_1^2 + \|W_1\|^2 + \bar{\delta}_2^2 \qquad (36)
\end{aligned}$$

Then, create actual controller as

$$\begin{aligned}
u &= -\frac{1}{\kappa_2(q_2)}(b_2\varpi_2 + \varpi_1 + \frac{1}{2}\varpi_2 + \hat{W}_2^T\phi_2(q_2) + \xi_1 \\
&\quad + c_2\xi_2 - \dot{\alpha}_2^f) \qquad (37)
\end{aligned}$$

and the adaptive law is constructed as

$$\dot{\hat{W}}_2 = \beta_2(\varpi_2\phi_2(q_2) + e_2\phi_2(q_2)) - \zeta_2\hat{W}_2 \qquad (38)$$

where $b_2$ and $\zeta_2$ are positive parameters.

Plugging (37) and (38) into (36), it leads to

$$\begin{aligned}
\dot{V} &\le -\sum_{i=1}^{2}b_i\varpi_i^2 + \sum_{i=1}^{2}\frac{\zeta_i}{\beta_i}\tilde{W}_i^T\hat{W}_i - k_2e_2^2 \\
&\quad + \frac{1}{2}\bar{\delta}_1^2 + \|W_1\|^2 + \bar{\delta}_2^2 \qquad (39)
\end{aligned}$$

## IV. Stability Analysis

**Theorem 1**: For the considered system (9) with the virtual controller (23), the actual controller (37), the adaptive laws (24), (38), the SPEM (31), and the command filter (12) under Assumption 1, all the internal states maintain boundness and the tracking error $z_1$ can reach an arbitrarily small set of zero. In addition, all states are constrained within the pre-designed set.

**Proof:** Based on Lemma 1, one has

$$\frac{\zeta_i}{\beta_i}\tilde{W}_i^T\hat{W}_i \le \frac{\zeta_i}{2\beta_i}W_i^TW_i - \frac{\zeta_i}{2\beta_i}\tilde{W}_i^T\tilde{W}_i \qquad (40)$$

By plugging (40) into (39), one can obtain

$$\dot{V} \le -\sum_{i=1}^{2} b_i \varpi_i^2 - \sum_{i=1}^{2} \frac{\zeta_i}{2\beta_i} \tilde{W}_i^T \tilde{W}_i - k_2 e_2^2$$

$$+ \frac{1}{2}\bar{\delta}_1^2 + \|W_1\|^2 + \sum_{i=1}^{2} \frac{\zeta_i}{2\beta_i} W_i^T W_i + \bar{\delta}_2^2$$

$$= -\acute{C}V + \acute{D} \tag{41}$$

where $\acute{C} = \min\{2b_1, 2b_2, \zeta_1, \zeta_2, 2k_2\}$, $\acute{D} = \frac{1}{2}\bar{\delta}_1^2 + \|W_1\|^2 + \sum_{i=1}^{2} \frac{\zeta_i}{2\beta_i} W_i^T W_i + \bar{\delta}_2^2$.

Next, we consider the convergence of the compensation signal, construct the following Lyapunov function:

$$V_\xi = \sum_{i=1}^{2} \frac{1}{2}\xi_i^2 \tag{42}$$

The time derivative of $V_\xi$ is

$$\dot{V}_\xi = \sum_{i=1}^{2} \xi_i \dot{\xi}_i$$

$$= -\sum_{i=1}^{2} c_i \xi_i^2 + \xi_1 \eta_2 \tag{43}$$

According to Lemma 1, we get

$$\xi_1 \eta_2 \le \frac{1}{2}\xi_1^2 + \frac{1}{2}\bar{\eta}_2^2 \quad |\eta_2| \le \bar{\eta}_2 \tag{44}$$

where $|\eta_2| \le \bar{\eta}_2$, then, substituting (44) into (43), one has

$$\dot{V}_\xi \le -(c_1 - \frac{1}{2})\xi_1^2 - c_2\xi_2^2 + \frac{1}{2}\bar{\eta}_2^2$$

$$\le -CV_\xi + D \tag{45}$$

where $C = \min\{2c_1 - 1, 2c_2\}$, $D = \frac{1}{2}\bar{\eta}_2^2$

According to (41) and (45), the errors $\varpi_i, \tilde{W}_i, e_2$ and the compensation signal $\xi_i$ $(i = 1, 2)$ in the considered system are bounded. Thus, the virtual signal $\alpha_1$, the input signal $u$ and adaptive law $\dot{\hat{W}}_i$ $(i = 1, 2)$ are also bounded. $\alpha_2^f$ is bounded because of the boundness of the filtering error $\eta_2$. From $\varpi_i = z_i - \xi_i$ $(i = 1, 2)$ and the boundness of $\varpi_i$ and $\xi_i$, the boundness of $z_i$ is derived. Meanwhile, the tracking error $z_1$ will incline to an arbitrarily small set of zero when the time tends to infinity. Since $z_1$ is bounded, we get that $q_1$ tracks up $y_r^*$. Then, it can be derived that $y$ follows $y_r$ based on the inverse mapping. Combining (11) with the boundness of $y_r^*$ and $\alpha_2^f$, $q_i$ $(i = 1, 2)$ is bounded. In addition, according to the one-to-one mapping, we can obtain the inequality $-a_i < x_i = \frac{2a_i}{\pi}\arctan(\frac{\pi q_i}{2a_i}) < a_i$. Therefore, the full-state constraints are implemented.

## V. SIMULATION

In this part, the simulation consequences are conducted to illustrate the effectiveness of the constructed control scheme. We introduce the ship autopilot parameters in [13]. The ship particulars are chosen as follows: draft $8.0m$, length between perpendiculars $L_{pp} = 126m$, block coefficient $0.681$, breath molded $B = 20.8m$, forward speed $7.72m/s$. Based on the

ship particulars, the autopilot mode parameters are selected as $a_1 = 1$, $a_2 = 0.6$, $T = 216$, $K = 0.478$. The reference heading trajectory is offered as $y_r = \sin(t)$.

For the convenience of simulation, the initial value of $[x_1(0), x_2(0), \xi_1(0), \xi_2(0)]^T$ are selected as $[0, 0, 0, 0]^T$. The value of the design parameters are selected as follows: $\tau_2 = 0.01$, $b_1 = 10$, $b_2 = 0.009$, $c_1 = 100$, $c_2 = 20$, $k_1 = 15$, $k_2 = 20$, $\beta_1 = 100000$, $\beta_2 = 800000$, $\zeta_1 = 10$, $\zeta_2 = 50$ and all the initial values of weights for the FLSs are zero. The state constraints are constructed as $a_1 = 4.5 + 0.21\sin(t)$, $a_2 = 4.5 + 0.21\sin(t)$.

The simulation curves are shown in Figs. 1-7. Specifically, Fig. 1 displays the system output trajectory $y$ and desire signal trajectory $y_r$, and the constraint functions $\pm a_1$. Fig. 2 are the trajectories of the system state $x_2$ and the constraint functions $\pm a_2$. According to Fig. 1-2, we can obtain that system output $x_1$ follows the desired signal $y_r$ well and states $x_1, x_2$ are constrained in specific sets, respectively. Fig. 3 displays the trajectories of the compensating signals $\xi_1$ and $\xi_2$. Fig. 4 plots the curve of system input $u$. The trajectories of $\|\hat{W}_1\|$ and $\|\hat{W}_2\|$ are shown in Fig. 5-6. The outcome of the composite adaptive fuzzy control is displayed in the Fig. 7. Based on Fig. 7, we can get that $\hat{W}_2^T \phi_2(q_2)$ can approximate function $F_2(q_2)$ well. Thus, it is not difficult to figure out that $\hat{f}_2(x_2) = \frac{\hat{F}_2(q_2)}{\kappa_2(q_2)} + p_2(q_2)$ can estimate the $f_2(x_2)$ well. In addition, from Fig. 1-6, it is easy to obtain that all the states in the considered system maintain boundness. Based on the above results, the effectiveness of the proposed control strategy is proved.

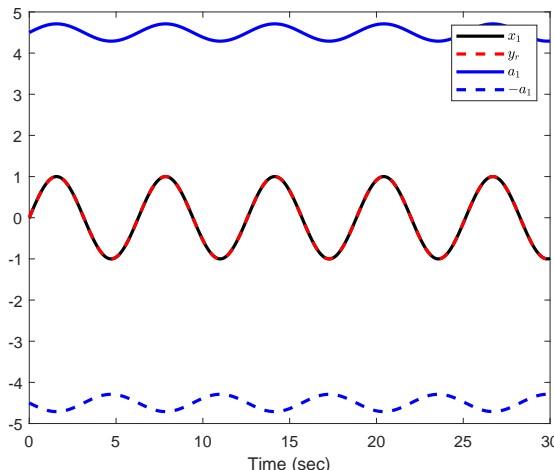

Fig. 1.   Trajectories of $x_1$, $y_r$ and $\pm a_1$.

## VI. CONCLUSION

In this work, a composite adaptive fuzzy control scheme is set up for intelligent ship autopilot with full-state constraints by utilizing the command-filter approach which is applied to dispose the complexity explosion problem. During the process of the controller design, FLSs are applied to dispose the unknown nonlinear terms. To tackle the issue of

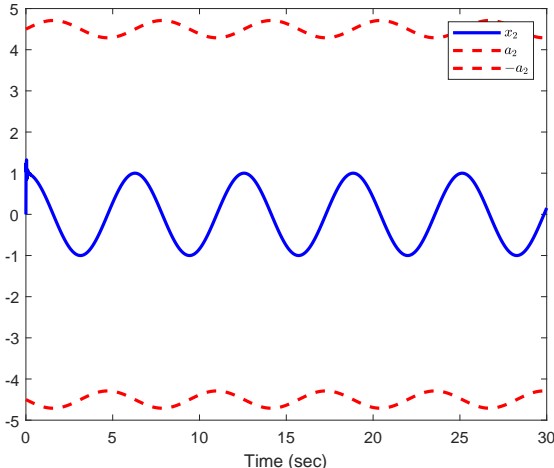

Fig. 2. Trajectories of $x_2$ and $\pm a_2$.

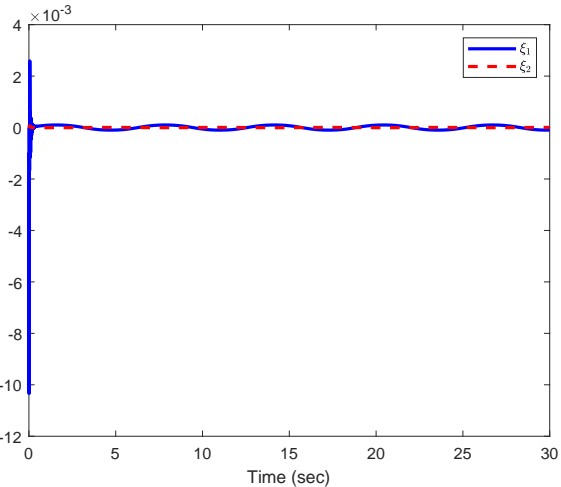

Fig. 3. Trajectories of $\xi_1$ and $\xi_2$.

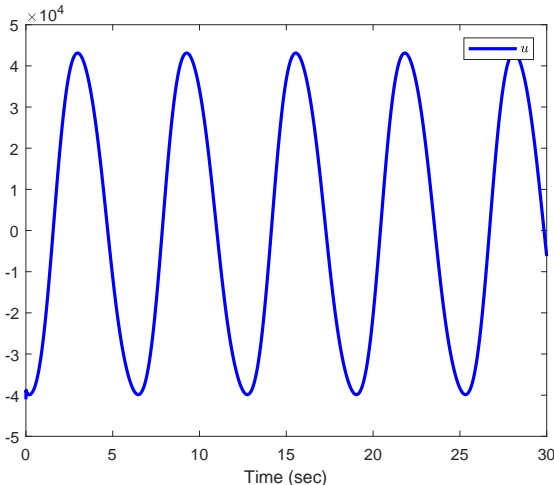

Fig. 4. The signal $u$.

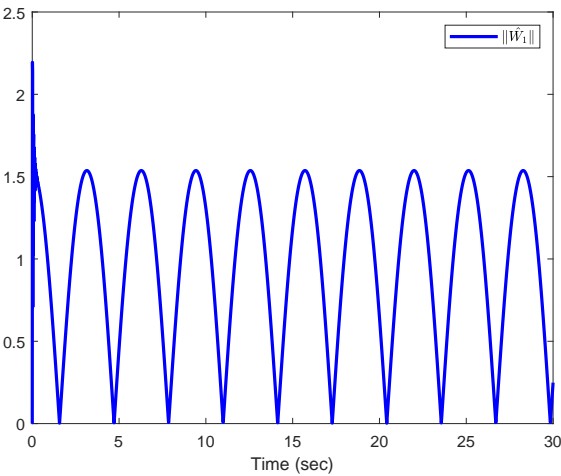

Fig. 5. Trajectory of $\|\hat{W}_1\|$.

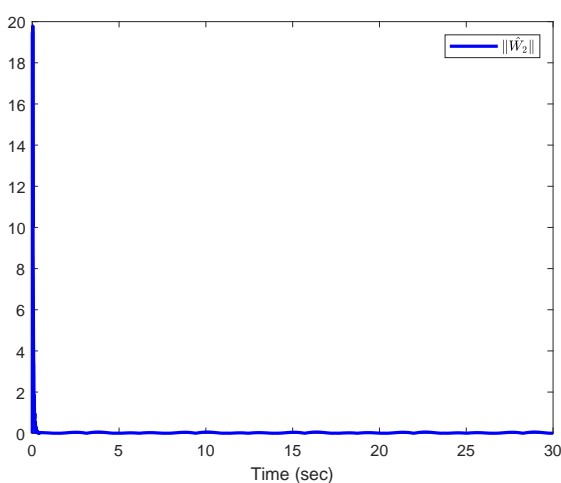

Fig. 6. Trajectory of $\|\hat{W}_2\|$.

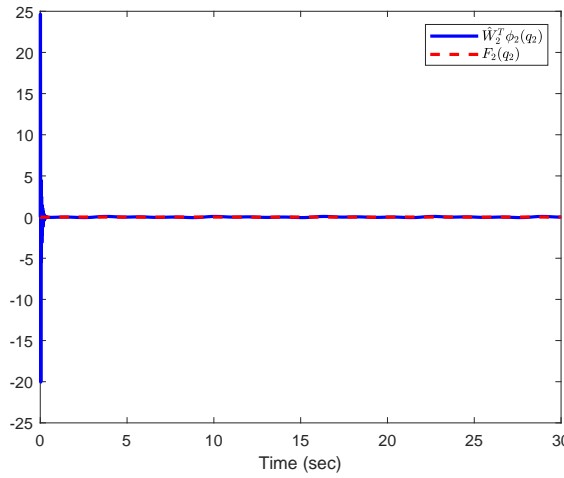

Fig. 7. Trajectories of $\hat{W}_2^T \phi(q_2)$ and $F_2(q_2)$.

nonlinear systems with full-state constraints, we introduce a tan-type nonlinear mapping approach. The SPEM can change the weights of FLSs and improve the approximate ability. The constructed control strategy can ensure all states in the considered system are bounded, and all signals satisfy the state constraints. Moreover, the output signal can track the desired signal. Finally, based on the simulation consequences, we can conclude the validity of the designed control strategy.

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
