# OpenReview forum: "Command Filter-based Adaptive Fuzzy Tracking Control for Intelligent Ship Autopilot with Full-state Constraints"
_IEEE.org/ICIST/2024/Conference — IEEE ICIST 2024 Conference Submission_

### Official Review · Reviewer_fy55 · 2024-08-26
**Accept**

**Rating:** 10
**Confidence:** 5

**Review:**

In this article, a command-filtered composite adaptive fuzzy tracking control problem is considered for intelligent ship autopilot with full-state constraints. The designed control strategy can ensure the boundedness of all states in the considered system
satisfies the state constraints. Overall, the language and organization are satisfactory. It can be accepted as a conference paper.

---

### Official Review · Reviewer_PHtT · 2024-08-29
**Accept after modification**

**Rating:** 6
**Confidence:** 4

**Review:**

1、It is recommended that the author revise the introduction to make it more logical and layered.
2、There are numerous grammatical problems in the text, and the author should scrutinize the entire text and make touch-ups and corrections to statements.
3、Definitions 1-3 are missing from the text and the authors are advised to check the article carefully.
4、Figure 6 cannot clearly express the trajectory. It is recommended that the author readjust the simulation diagram.
5、Is “F2 (q2)” in Fig. 7 always 0 during the simulation? The authors should provide a clear explanation of the lines in the figure.

---

### Official Review · Reviewer_nkT9 · 2024-08-30
**This paper can be accepted**

**Rating:** 7
**Confidence:** 3

**Review:**

1.	Although the introduction section of the article mentions the application of adaptive backstepping, Lyapunov analysis method, and fuzzy logic systems in nonlinear system control, the description of these methods is relatively brief. It is suggested to further expand the elaboration in the theoretical foundation section, introduce the basic ideas of these methods in detail, and add a review of the latest literature in related fields. Additionally, provide a detailed description of the research motivation in the introduction section, explaining the necessity of studying the control problem of intelligent ship autopilots under full-state constraints and the deficiencies of existing methods in addressing such problems. This will help readers better understand the importance and urgency of the research.

2.	When introducing the mathematical model of ship course control, although some parameters (such as a1, a2, T, K, etc.) are explained, there is a lack of detailed explanation of the actual physical meaning of these parameters and their impact on system behavior. It is necessary to add detailed explanations of the model parameters to enhance readers' understanding of the validity of the model.

3.	Although the controller design section provides detailed steps and formulas, the explanation of some key steps and formula derivation processes is insufficient. It is necessary to add detailed derivations of key steps and formulas to help readers better follow and understand the design ideas of the controller. For example, when introducing tan-type nonlinear mapping and SPEM, it is necessary to specifically elaborate on how they help solve the full-state constraint problem and enhance the approximation capability of the fuzzy logic system.

4.	While the simulation section shows results such as system output, state variables, and control inputs, there is a lack of deep analysis of the simulation results. It is necessary to conduct an in-depth discussion of the simulation results, including verification of the effectiveness of the control strategy, quantitative evaluation of system performance, and possible directions for improvement.

5.	Provide more detailed captions and explanations for the charts in the article. Below the charts showing simulation results, briefly describe the key information and data trends in the charts. This will help readers quickly understand the simulation results and evaluate the effectiveness of the proposed method.

6.	In the conclusion section, apart from summarizing the main research findings, it is also necessary to discuss the limitations of the research and future research directions. For example, you can discuss how to apply this method to other types of nonlinear systems or real-world scenarios, or point out issues and challenges that require further investigation. At the same time, you can also discuss the potential advantages and limitations of this method in practical applications.

---

### Decision · Program_Chairs · 2024-09-06

Accept (Oral)